
# The Pacific-Indian Ocean Associated Mode in CMIP5 Models

Minghao Yang, Xin Li*, Weilai Shi, Chao Zhang, Jianqi Zhang

College of Meteorology and Oceanography, National University of Defense Technology, Nanjing, 211101, China

*Correspondence to*: Xin Li (lixin_atocean@sina.cn)

**Abstract.** The Pacific-Indian Ocean associated mode (PIOAM) is the product of the tropical air-sea interaction at the cross-basin scale and the main mode of ocean variation in the tropics. Evaluating the capability of current climate models to simulate the PIOAM and finding the possible factors that affect the simulation results are beneficial to obtain more accurate future climate change prediction. Based on 55-yr the Hadley Centre Global Sea Ice and Sea Surface Temperature (HadISST) reanalysis and the output data from twenty-one Coupled Model Intercomparison Project (CMIP) phase 5 (CMIP5) models, the PIOAM in these CMIP5 models is assessed. It is found that the explained variance of PIOAM in almost all twenty-one CMIP5 models are underestimated. Although all models reproduce the spatial pattern of the positive sea surface temperature anomaly in the eastern equatorial Pacific well, only one-third of these models successfully simulate the ENSO mode with the east-west inverse phase in the Pacific Ocean. In general, CCSM4, GFDL-ESM2M and CMCC-CMS have a stronger capability to capture the PIOAM than that of the other models. The strengths of the PIOAM in the positive phase in less than one-fifth of the models are slightly stronger, and very close to HadISST reanalysis, especially in CCSM4. The interannual variation of PIOAM can be measured by CCSM4, GISS-E2-R and FGOALS-s2. Further analysis indicates that considering the carbon cycle, resolving stratosphere, chemical process or increasing the horizontal resolution of the atmospheric model may effectively improve the performance of the model to simulate the PIOAM.

## 1. Introduction

As early as the 1960s, Bjerkness (1966, 1969) studied the phenomenon of El Niño-Southern Oscillation (ENSO). Since then, the impact of ENSO on global climate has become a major concern in climate research. ENSO in the Pacific Ocean is the strongest interannual signal of global climate change, and has been extensively studied by a large number of scholars, including its occurrence and development

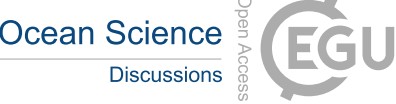

mechanism (Wyrtki, 1975; Philander et al., 1984; Suarez and Schopf, 1988; Jin, 1997; Li and Mu, 1999;
Li and Mu, 2000; Li, 2002), its evolution characteristics and its impact on global weather and climate
(Bjerknes, 1966; Rasmusson and Wallace, 1983; Ropelewski and Halpert, 1987; Li, 1990; Webster and
Yang, 1992; Zhou and Zeng, 2001; Mu and Duan, 2003; Mu et al., 2007; Zheng et al., 2007). At the end
of the 20th century, an interannual climate anomaly characterized by a sea surface temperature anomaly
(SSTA) of opposing sign in the western and eastern tropical Indian Ocean, known as the Indian Ocean
dipole (IOD), was reported by Saji et al. (1999) and Webster et al. (1999) and was catalogued as one of
the major ocean-atmosphere coupled phenomena. The SSTA in the tropical Indian Ocean subsequently
has been widely studied, and a great deal of literature has discussed the causes and mechanisms of the
IOD, as well as its weather and climate impacts (Li and Mu, 2001; Li et al., 2003; Saji and Yamagata,
2003; Cai et al., 2005; Rao et al., 2007; Zheng et al., 2013; Wang and Wang, 2014).
IOD was initially thought to be generated only by independent air-sea interactions in the tropical
Indian Ocean, but some studies have suggested that the tropical Indian Ocean SSTA in 1997/1998 was
caused by the influence of the ENSO event in the Pacific Ocean on the surface wind field of the Indian
Ocean through anti-walker circulation over the equator, thus causing the SSTA in the Indian Ocean (Yu
and Rienecker, 1999). It has also been suggested that the east-west asymmetry anomaly of the Indian
Ocean SSTA in 1997/1998 may contain the triggering process of ENSO (Ueda and Matsumoto, 2000).
Li et al. (2002) showed that there is a significant negative correlation between the tropical Indian Ocean
SSTA dipole event and the Pacific SSTA dipole event (similar to ENSO mode) using statistical analysis.
Huang and Kinter (2002) also noted that there was a significant relationship between IOD in the Indian
Ocean and ENSO in the Pacific Ocean.
The movements and changes of Earth's fluids (atmosphere and oceans) have a certain connection,
and the change in tropical sea surface temperature (SST) should not be an isolated phenomenon. IOD in
the Indian Ocean and ENSO in the Pacific Ocean, both as significant basin-scale signals, are supposed
to be closely related and interact with each other. Although the type of relationship between ENSO and
IOD has not yet been fully demonstrated, extensive research has shown that both SST and the air-sea
systems in the Pacific Ocean and the Indian Ocean are closely linked (Klein and Soden, 1999; Li et al.,
2008; Huang and Kinter, 2002; Li et al., 2003; Annamalai et al., 2005; Cai et al., 2019). The walker
circulation anomaly induced by SSTA over the equatorial Pacific Ocean will cause a walker circulation



anomaly over the Indian Ocean, which could inspire the occurrence and development of IOD in the
Indian Ocean driven by abnormal wind stress in the lower layer. On the other hand, Indonesian
Throughflow also plays a role in the connection between ENSO and IOD. The cold (El Nino) or warm
(La Nina) SST of the warm pool in the Pacific Ocean can cool or warm the SST in the eastern equatorial
Indian Ocean through the Indonesian Throughflow, which is conducive to the establishment of a positive
or negative phase of IOD.
Yang and Li (2005) found the first leading mode of the tropical Pacific-Indian SSTA reflecting the
opposite phase characteristics of both the middle west Indian Ocean and equatorial middle east Pacific
Ocean and both the eastern Indian Ocean and equatorial western Pacific Ocean, from which they
proposed the concept of the Pacific-Indian Ocean associated mode (PIOAM), and noted that the PIOAM
can better reflect the influence of the tropical SSTA on Asian atmospheric circulation. Yang et al. (2006)
subsequently found that the influences of the PIOAM and the ENSO model on summer precipitation and
climate in China were very different, and their numerical experiments also showed that the simulation
results obtained by considering the PIOAM were more consistent with observation data. By analyzing
the monthly thermocline temperature anomaly (TOTA) from 1958-2007 and the weekly sea surface
height (SSH) anomaly from 1992-2011 in the tropical Pacific–Indian Ocean, Li et al. (2013) further found
that the PIOAM are more obviously in the subsurface ocean temperature anomaly field, especially in the
thermocline. Based on the simulation results of the LASG/IAP (State Key Laboratory of Numerical
Modeling for Atmospheric Sciences and Geophysical Fluid Dynamics/Institute of Atmospheric Physics)
Climate system Ocean Model (LICOM), version 2 (LICOM2.0) (Liu et al. 2012) and observation data,
Li and Li (2017) proved that PIOAM is an important tropical Pacific-Indian Ocean SST variation mode
that actually exists both in observation and simulation. Therefore, when studying the influence of SSTA
in the Pacific and Indian oceans on weather and climate, the Pacific and Indian oceans should be
considered as unified. Since the PIOAM is so important, how well do current climate models simulate
it? To answer this question, the outputs from the climate system models for the Coupled Model
Intercomparison Project (CMIP) phase 5 (CMIP5) were used for this research, from which we aim to
provide a more complete evaluation of the PIOAM and try to find possible reasons that cause the
simulation biases. In the following, Sect. 2 includes a brief description of the reanalysis dataset, CMIP5
models, and the methods used in this study. Section 3 presents the assessments of the PIOAM in the



CMIP5 models. A conclusion and discussion are given in Sect. 4.

**2. Data and methods**

The reanalysis data from the Hadley Centre Global Sea Ice and Sea Surface Temperature (HadISST)
(Rayner et al., 2003) data set is used for this study. The data are monthly averaged data from 1951 to
2005 with a spatial resolution of 1°×1°. Brief information for the 21 CMIP5 models used in this article
is provided in Table 1. Considering that output data resolutions vary between the models, we first
interpolated all data into a 1°×1° grid to facilitate comparison between the models and HadISST
reanalysis.

Table 1. List of 21 selected CMIP5 climate models.

| Model name | Modeling group | Oceanic resolution (lon×lat) |
|---|---|---|
| CanESM2 (Second Generation Canadian Earth System Model) | Canadian Centre for Climate Modeling and Analysis, Canada | 256×192 |
| CCSM4 (The Community Climate System Model, version 4) | NCAR, USA | 320×384 |
| CMCC-CESM (Centro Euro-Mediterraneo sui Cambiamenti Climatici (CMCC) Carbon Earth System Model) | CMCC, Italy | 182×149 |
| CMCC-CM (CMCC Climate Model) | CMCC, Italy | 182×149 |
| CMCC-CMS (CMCC-CM with a resolved stratosphere) | CMCC, Italy | 182×149 |
| CNRM-CM5 (Centre National de Recherches Météorologiques (CNRM) Coupled Global Climate Model, version 5) | CNRM, France | 362×292 |
| FGOALS-s2 (The Flexible Global Ocean-Atmosphere-Land System model, Spectral Version 2) | LASG, China | 360×196 |
| GFDL-ESM2M (Earth System Model of Geophysical Fluid Dynamics Laboratory (GFDL) with Modular Ocean Model, version 4) | GFDL, USA | 144×90 |
| GISS-E2-H (Goddard Institute for Space Studies (GISS) Model E version 2 (GISS-E2) with HYCOM ocean model) | NASA, USA | 144×90 |
| GISS-E2-H-CC (GISS-E2-H with carbon cycle) | NASA, USA | 144×90 |
| GISS-E2-R (GISS-E2 with Russell ocean model) | NASA, USA | 144×90 |
| GISS-E2-R-CC (GISS-E2-R with carbon cycle) | NASA, USA | 144×90 |
| HadCM3 (the third version of he Hadley Centre coupled model) | Met Office Hadley Centre, UK | 288×144 |
| HadGEM2-AO (Hadley Global Environment Model 2 (HadGEM2)-Atmosphere-Ocean) | Met Office Hadley Centre, UK | 360×216 |
| HadGEM2-CC (HadGEM2-Carbon Cycle) | Met Office Hadley Centre, UK | 360×216 |
| HadGEM2-ES (HadGEM2-Earth System) | Met Office Hadley Centre, UK | 360×216 |
| IPSL-CM5B-LR (Institut Pierre Simon Laplace Climate Model 5B (LPSL-CM5B)-Low Resolution) | IPSL, France | 182×149 |
| IPSL-CM5B-MR (LPSL-CM5B 5A-Medium Resolution) | IPSL, France | 182×149 |



| MIROC-ESM (Model for Interdisciplinary Research on Climate, Earth System Model) | Atmosphere and Ocean Research Institute (AORI), Japan | 256×192 |
|---|---|---|
| MIROC-ESM-CHEM (An atmospheric chemistry coupled version of MIROC-ESM) | AORI, Japan | 256×192 |
| NorESM1-M (Norwegian Climate Centre Earth System Model) | Norwegian Climate Centre, Norway | 384×320 |


The PIOAM is determined according to the method of Ju et al. (2004) and Li et al. (2018), that is,
the first leading mode of the tropical Pacific-Indian ocean SSTA is used to represent the PIOAM. Ju et
al. (2004) used this method to analyze SSTA in the tropical Pacific-Indian Ocean in different seasons,
and found the existence of PIOAM in all seasons with a contribution to total variance of more than 33%,
indicating that the spatial distribution structure of PIOAM was stable.
Accounting for the intimate connection between the Pacific ENSO mode and the Indian Ocean
dipole, Yang et al. (2006) argued that the PIOAM index (PIOAMI) can be defined as the respectively
normalized east-west SSTA differences of the equatorial areas in the two oceans. As to the SSTA, the
SSTA of ENSO is stronger than that in the equatorial Indian Ocean because of the larger Pacific basin;
however, as to the influence of the SSTA on East Asia, a series of numerical experiments clearly indicate
that the effect of SSTA forcing on the Indian Ocean is stronger than that of the eastern equatorial Pacific
(Shen et al., 2001; Guo et al., 2002; Guo et al., 2004; Yang et al., 2006). Therefore, the PIOAMI is
defined on the basis of the respective normalized dipoles in the Pacific and the Indian Ocean. According
to the method of Yang et al. (2006), The PIOAMI is defined as follows:
$PIOAMI = IOI + POI$ (1)
$IOI = SSTA(5° - 10° \text{N}, \ 50° - 65° \text{E}) - SSTA(10° \text{S} - 5° \text{N}, 85° - 100° \text{E})$ (2)
$POI = SSTA(5° \text{S} - 5° \text{N}, \ 130° - 80° \text{E}) - SSTA(5° \text{S} - 10° \text{N}, 140° - 160° \text{E})$ (3)
where IOI and POI are the normalized Indian Ocean and Pacific Ocean indeices, respectively.

**3. Results**

**3.1 Spatial pattern**

Figure 1 depicts the spatial pattern of PIOAM in the selected 21 CMIP5 models and their differences
compared to HadISST reanalysis (Fig. 1a). Figure 1.b shows the results of a multi-model ensemble
(MME) that represents the mean of the results from all selected models. To better and objectively evaluate





the capability of each model in simulating PIOAM, a Taylor diagram (Fig. 2) is also adopted to concisely
display the relative information from multiple models, so that the differences among the simulations from
all models are revealed clearly (Taylor, 2001; Jiang and Tian, 2013; Yang et al., 2018). According to
HadISST reanalysis (Fig. 1a), with a 47% contribution to total variance, the PIOAM has a warm tongue
spatial pattern in the eastern equatorial Pacific Ocean, whereas there is negative SSTA in the western
equatorial Pacific Ocean, which exhibits an obvious ENSO mode in the Pacific Ocean. In addition, there
are obvious positive SSTA in the western Indian Ocean region of the PIOAM, but the SSTA in the eastern
equatorial Indian Ocean region remain positive. Considering that the so-called IOD is defined by the
difference between the SSTA in the western equatorial Indian Ocean and that in the eastern equatorial
Indian Ocean, this indicates zonal heat contrast of the Indian Ocean SSTA. Although it is called a dipole,
it is not related to the mathematic meaning (Li et al., 2002; Yang et al., 2006). Therefore, it can be
considered that the PIOAM presents an IOD mode in the Indian Ocean region.
Figure 1 shows that all of these models can generally reproduce the spatial pattern of PIOAM, yet
large discrepancies exist regarding the strength, and the differences between the models are also
significant. Except for the contribution to total variance of PIOAM in CCSM4 and CMCC-CESM are
nearly consistent with HadISST reanalysis, the variance contribution of PIOAM in almost all CMIP5
models are lower than those in the HadISST reanalysis, especially CMCC-CM with a contribution to
total variance as small as 26%. In terms of strength, it is apparent that the simulation errors of these
models are mainly concentrated in the Pacific Ocean compared to the Indian Ocean. Compared to the
HadISST reanalysis, a majority of models overestimate the strength of PIOAM in the equatorial east
Pacific and central Pacific; only one-seventh of the models (IPSL-CM5A-LR, IPSL-CM5A-MR and
MIROC-ESM) underestimate the strength of PIOAM in the equatorial east Pacific, while the simulation
results of HadGEM2-AO and CMCC-CM in the equatorial central Pacific and western Pacific are weak.
The simulation errors of the strength of the ENSO mode in CCSM4, CMCC-CMS, GFDL-ESM2M and
GISS-E2-R-CC are lower than those in other models. For the Indian Ocean, the strengths of PIOAM in
only approximately one-quarter of the models (CanESM2, CMCC-CESM, GISS-E2-H-CC, HadCM3
and HadGEM2-AO) are basically consistent with HadISST reanalysis with small simulation errors.
Nearly half of the models were smaller for the eastern Indian Ocean, whereas more than half were larger
for the western Indian Ocean. In general, the simulation error in the Indian Ocean region is significantly



smaller than that in the Pacific region. According to Fig. 2, it is apparent that the root mean square errors
(RMSEs) in MIROC-ESM-CHEM, IPSL-CM5A-LR and MIROC-ESM are relatively large, which
means that the capabilities of these modes to simulate the strength of PIOAM are still inadequate,
whereas the RMSEs in CCSM4, CMCC-CMS and GFDL-ESM2M are smaller than in other models with
a better performance. In addition, as shown in Fig. 1.b, MME better simulates the strength of PIOAM in
in the Indian Ocean region, but the simulation errors in the equatorial Pacific region are larger than that
of the HadISST reanalysis.

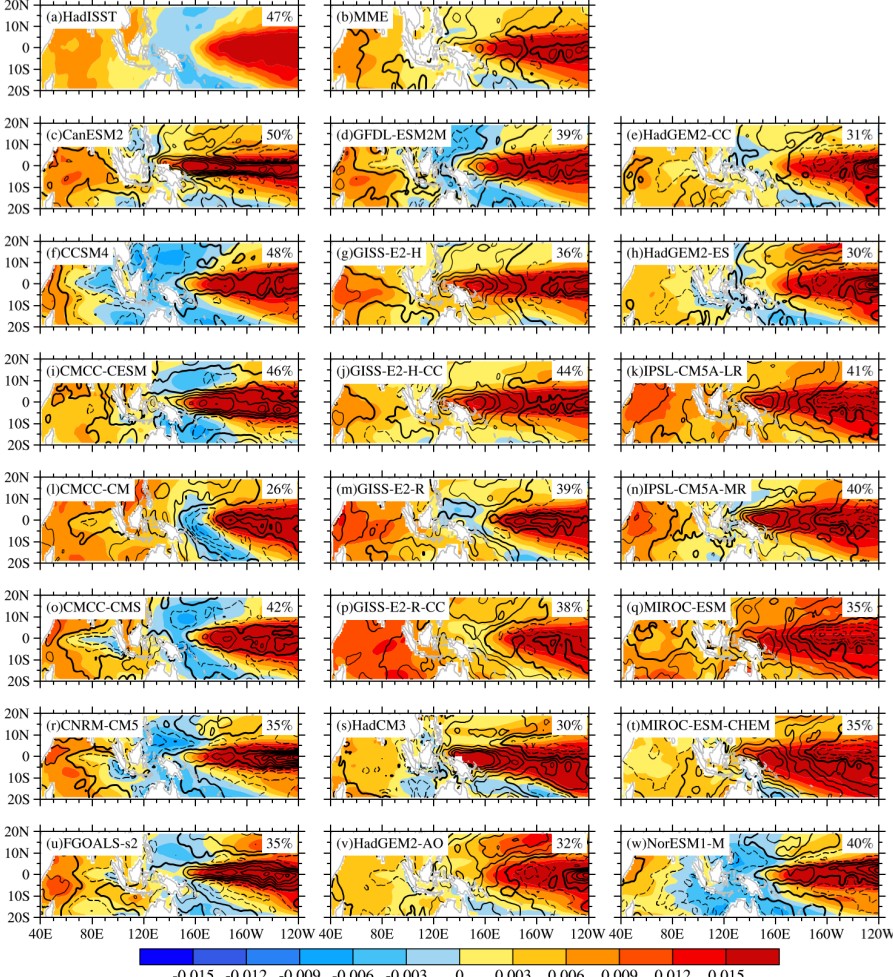


**Figure 1: PIOAM (shading) and the difference between each model and HadISST reanalysis (contour, with**

**an interval of 0.003, shown as black bold lines represent the contour with the zero value, unit: ℃).**



As for spatial patterns, the IOD mode in the Indian Ocean region can be simulated in almost all
models except MIROC-ESM-CHEM. Although all these models reproduce the spatial pattern of the
positive SSTA well in the eastern equatorial Pacific, only one-third of the models (CCSM4, CMCC-CM,
CMCC-CMS, CNRM-CM5, FGOALS-s2, GFDL-ESM2M and NorESM1-M) successfully simulate the
ENSO mode with the east-west inverse phase in the Pacific Ocean. In addition, the simulated positive
SSTAs in the eastern equatorial Pacific in HadCM3 and MIROC-ESM-CHEM are further south.
According to Fig. 2, more than one-third of these models (CCSM4, CMCC-CMS and GFDL-ESM2M,
etc.) can simulate the spatial pattern of PIOAM well, and the spatial correlation coefficients between
these models and the HadISST reanalysis are all greater than 0.9, especially CCSM4, which is as high as
0.95. In contrast, the spatial pattern of PIOAM in MIROC-ESM-CHEM is unsatisfactory with a spatial
correlation coefficient of only 0.69. The simulation results of HadCM3 and MIROC-ESM are also
relatively poor, and the spatial correlation coefficients with HadISST reanalysis are less than 0.8. It can
also be learned from Fig. 2 that, for the standard deviation of PIOAM, very large differences exist among
these models. The standard deviations of PIOAM in IPSL-CM5A-LR, MIROC-ESM and GISS-E2-R-
CC are quite different from those of the HadISST reanalysis, while the simulation results of CMCC-
CMS, GFDL-ESM2M and HadGEM2-CC are basically close to those of the HadISST reanalysis and
have better performance. It is noteworthy that the standard deviations of PIOAM in more than half of
these models are smaller than that of the reanalysis, and their differences are large. Although the spatial
pattern of PIOAM in MME is closer to the HadISST reanalysis and the RMSE is smaller than the vast
majority of single models, the standard deviation of PIOAM in MME is smaller than that of the HadISST
reanalysis.
In general, CCSM4, GFDL-ESM2M and CMCC-CMS have a stronger ability to simulate the
PIOAM. In addition, although the MME of these models may not be as good as that of a single model in
some specific aspects, overall, considering spatial pattern, standard deviation and RMSE, MME is still
superior to most single models.

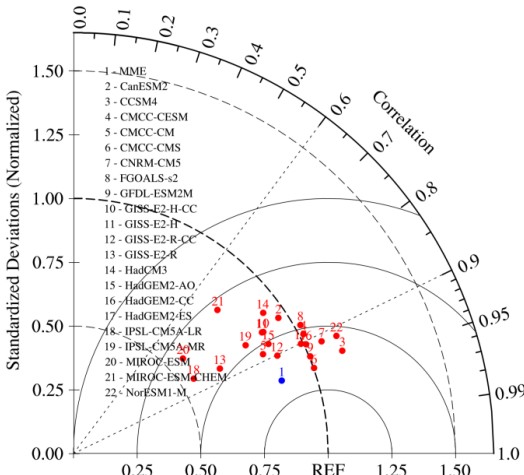

**Figure 2: Taylor diagrams of PIOAM.**

To further evaluate the differences between these models, Fig. 3 shows the distribution of standard deviations between the CMIP5 models, which clearly reflects the regional differences between the models. It is apparent that the differences are mainly concentrated in the eastern equatorial Pacific. Therefore, the emphasis of improving the model on simulating the PIOAM is to improve the capability of the model to simulate to the Eastern-Pacific (EP) type ENSO.

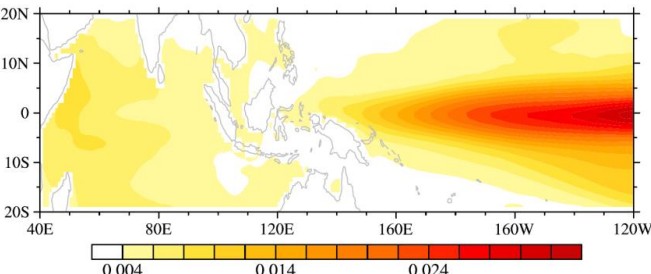

**Figure 3: The standard deviations of simulated PIOAM between the selected 21 CMIP5 models.**

**3.2 PIOAM index**

**3.2.1 Time series**

Figure 4. shows the monthly time series of the PIOAM index (PIOAMI), Pacific Ocean index (POI) and Indian Ocean index (IOI) from 1951 to 2005. The wavelet analysis of PIOAMI indicates that PIOAM



has obvious seasonal and interannual variations, as well as interdecadal variations (feature is omitted).
According to Fig. 4, POI and IOI have the same variation tendency at most times, thus the PIOAMI
amplitude is greatly enhanced. However, there are a few cases where the two change in opposing ways,
resulting in a much weaker PIOAMI. Moreover, from the time-series of PIOAMI, there is an interannual
oscillation of positive and negative phases in the PIOAM, and there is also a phenomenon that the
PIOAMI is very weak or not obvious in some years.

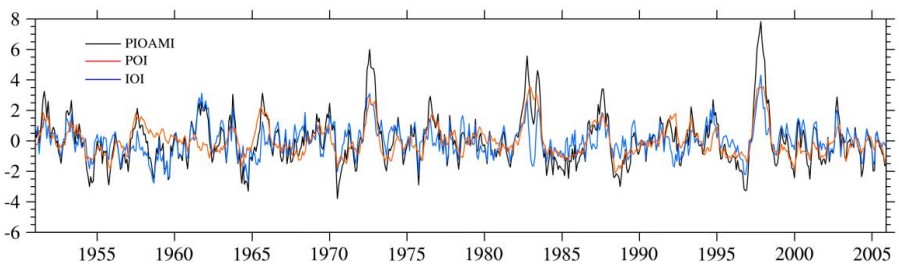

**Figure 4: Time-series of PIOAM index (black), Pacific Ocean index (red) and Indian Ocean index (blue) in**
**the HadISST reanalysis.**

Considering that the PIOAM mainly reaches its peak in autumn, we select the year with significant
positive and negative phases of PIOAM according to one standard deviation of PIOAMI in autumn, and
calculate the difference in autumn PIOAMI between each model and the HadISST reanalysis (see Fig. 5)
to further reveal the simulation of the CMIP5 models on the strength of the PIOAM. As shown by Fig.
5.a, the simulated strengths of the PIOAM in the positive phase are underestimated in most models,
whereas they are slightly overestimated in less than one-fifth of the models (CCSM4, CMCC-CMS,
CNRM-CM5 and GFDL-ESM2M) are slightly stronger, which are very close to the HadISST reanalysis,
especially in CCSM4. However, nearly half of the models overestimate the strength of the PIOAM in the
negative phase (Fig. 5.b), in which the simulation results of CanESM2 and GISS-E2-R are consistent
with the HadISST reanalysis. Although CCSM4 has a better performance in simulating the strength of
the PIOAM in the positive phase than other models, the simulation error of the negative phase is very
large.

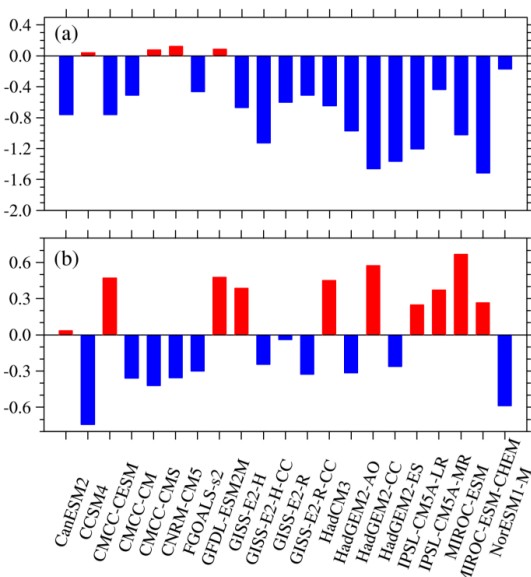


**Figure 5: Difference in the amplitude of the PIOAMI in the positive phase (a) and negative phase (b) between**
**the CMIP5 models and HadISST reanalysis.**

According to PIOAM positive and negative phase year based on the autumn PIOAMI, SSTAs in

the tropical Pacific-Indian Ocean in October are composed to obtain the spatial pattern of SSTAs in the

PIOAM positive and negative phases. It is clear in Fig. 6 that the SSTAs in the Pacific-Indian Ocean in

both the MME of CMIP5 models and HadISST reanalysis present patterns with a tripole structure, where

the Indian Ocean is represented by the IOD mode and the Pacific Ocean by the ENSO mode, which again

demonstrates the authenticity of PIOAM and the rationality of PIOAMI used in this article.

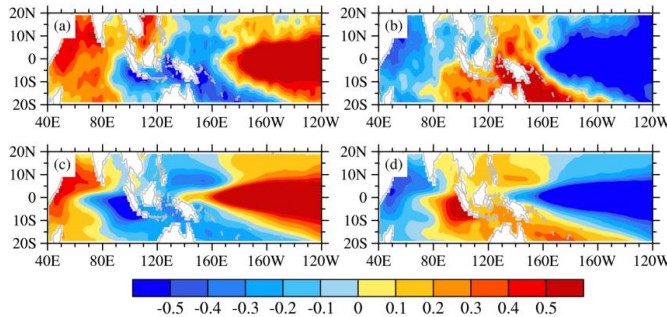

**Figure 6: The tropical Pacific-Indian Ocean SSTAs of the PIOAM positive (a, c) and negative (b, d) phase in**
**October in the HadISST reanalysis (a, b) and the MME of CMIP5 models (c, d) (unit: ℃).**





### 3.2.2 Interannual Variation of PIOAM

To evaluate the ability of these CMIP5 models to simulate the interannual variation of PIOAM, Fig. 7 shows the ratios of standard deviation of IOI, POI and PIOAMI in autumn in each model to those in the HadISST reanalysis. It can be found that the difference in the simulation results of the interannual variation of PIOAMI among these models is smaller compared to IOI and POI. The simulation results of CCSM4, GISS-E2-R and FGOALS-s2 are almost consistent with HadISST reanalysis, indicating that these three models have relatively strong capabilities to simulate the interannual variation of PIOAM. Except that NorESM1-M overestimates the interannual variation of PIOAM, the simulation results in most of the models are weak, especially MIROC-ESM, which leads to MME underestimating the interannual variation of PIOAM compared to the HadISST reanalysis. In addition, the interannual variations of IOI in GFDL-ESM2M, GISS-E2-R-CC and CMCC-CM are better than other models, whereas the simulation results are underestimated in most models. In contrast to IOI, the vast majority of models overestimate the interannual variations of POI, and the simulated interannual variations of POI in only three models (IPSL-CM5A-MR, CMCC-CESM and IPSL-CM5A-LR) are weaker than the HadISST reanalysis. Based on the above analysis, it is apparent that the interannual variation of PIOAM is more closely to IOI than POI, and the interannual variation of PIOAM in autumn can be measured by CCSM4, GISS-E2-R and FGOALS-s2.

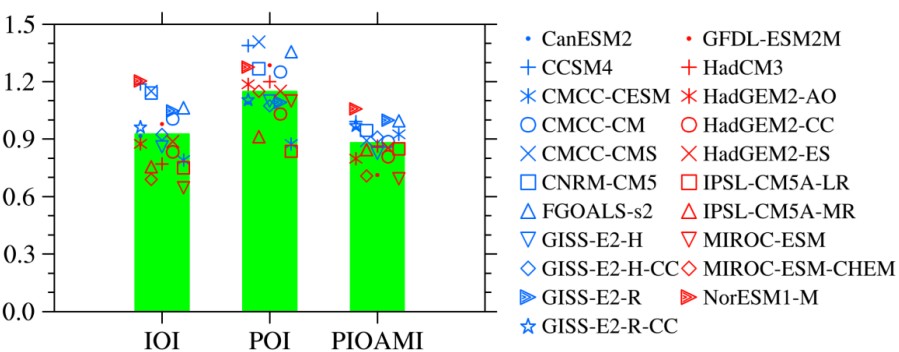

**Figure 7: Ratios of standard deviation of autumn IOI, POI and PIOAMI in each model to those in the HadISST reanalysis. Green bar represents the MME of the corresponding index.**

### 3.2.3 The relationship of PIOAM with ENSO and IOD

The lag-lead correlation analysis between PIOAMI and the Niño3.4 index derived from the





HadISST reanalysis shows that PIOAM has a close correlation with the ENSO mode at the same period
and one month lagging with the correlation coefficient of 0.68 (Fig. 8.a). In addition, PIOAM and IOD
also have a close correlation in the same period, with a correlation coefficient of 0.73 (Fig. 8.b), indicating
that the PIOAM can reflect the activities of ENSO in the Pacific Ocean and IOD in the Indian Ocean to
a considerable extent. In these CMIP5 models, more than one-half of the models successfully reproduce
the maximum correlation between PIOAM and ENSO in the same period. The correlation coefficients
of the PIOAMI and the Niño3.4 index in HadGEM2-AO and HadGEM2-ES are both 0.68, which is
consistent with the HadISST reanalysis, and the correlation coefficients of FGOALS-s2, GISS-E2-H and
GISS-E2-H-CC are 0.69, 0.69 and 0.70, respectively. However, the correlation coefficients of MIROC-
ESM and MIROC-ESM-CHEM are only 0.37 and 0.30, which are significantly different from the results
of the HadISST reanalysis and other models, indicating that the two models cannot simulate the close
relationship between the PIOAM and ENSO. In addition, the correlation coefficient of PIOAMI and
the Niño3.4 index in MME is 0.66, which is slightly lower than the HadISST reanalysis but shows the
close contemporaneity correlation between the PIOAM and ENSO; the overall change of the correlation
coefficient series is very close to the HadISST reanalysis.

For the relationship between the PIOAM and IOD, it is apparent from the HadISST reanalysis in

Fig. 8.b that the PIOAM and IOD show obvious close correlation in the same period, and the correlation
coefficient is as high as 0.73. It is satisfactory that all selected CMIP5 models successfully reproduce the
correlation between PIOAM and IOD in the same period, but the simulation results in more than half of
them are underestimated. Among these models, the simulation results of HadGEM2-ES and GIS-E2-R-
CC are basically consistent with the HadISST reanalysis, which shows that the two models have stronger
capability to simulate the relationship between PIOAM and IOD.

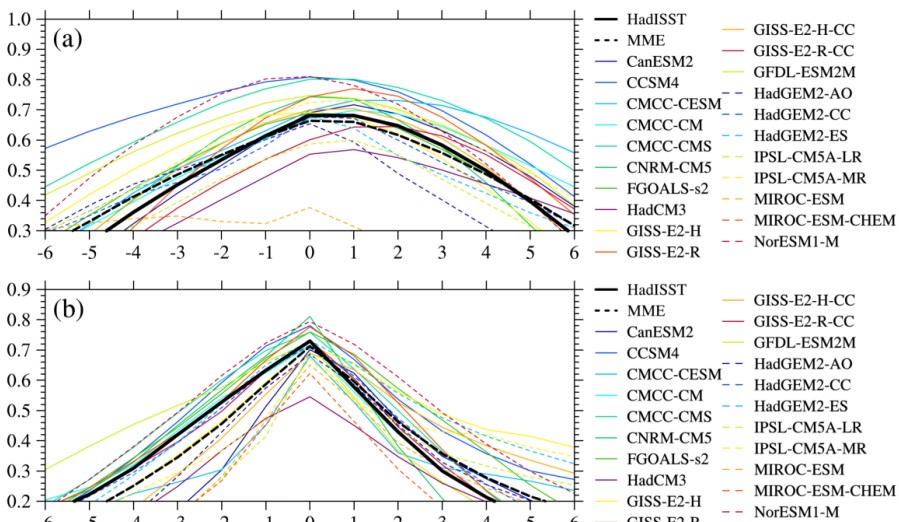

**Figure 8: The lag-lead correlation coefficient of the PIOAMI with the Niño3.4 index (a) and IOD index (b).**
**Ordinate represents the correlation coefficient, and abscissa is the lag in months: positive (negative) for the**
**Niño3.4 index or IOD index (PIOAMI) leading PIOAMI (Niño3.4 index or IOD index)**

**4. Possible causes of simulation errors**

It is undoubtedly difficult to directly find the factors that influence the model to simulate the PIOAM.

However, the simulation results of model families, such as CMCC, IPSL, MIROC, GISS and HadGEM2,

provide comparative data to find the possible reasons that may lead to simulation differences. Using the

CMCC model family as an example, CMCC-CM is a climate model. Because of this, CMCC-CM,

CMCC-CESM and CMCC-CMS consider more complete physical processes and closer to the real world.

CMCC-CESM is a carbon earth system model, and CMCC-CMS is a climate model with a resolved

stratosphere. It can be found from Fig. 1.i, l and o that although CMCC-CM, CMCC-CESM and CMCC-

CMS all overestimate the strength of the PIOAM in the equatorial eastern Pacific Ocean, CMCC-CESM

also overestimates it in western Pacific Ocean, but CMCC-CM is weak. While the simulation results in

CMCC-CMS in the Pacific Ocean, especially the equatorial western Pacific Ocean, are better than that

in CMCC-CM and CMCC-CESM, but the strength of the PIOAM in the East Indian Ocean is weak in

CMCC-CMS. In addition, the variance contribution of PIOAM in CMCC-CESM is very close to that in

the HadISST reanalysis, reaching 46%, followed by CMCC-CMS with 42%, while CMCC-CM only has

26%. As shown in Fig. 2, not only is the spatial pattern of PIOAM in CMCC-CMS better with a higher

spatial correlation coefficient, the RMSE is smaller and the standard deviation of PIOAM is basically





consistent with the HadISST reanalysis. It can be found that considering the carbon cycle or resolving
the stratosphere can effectively improve the capability to simulate the PIOAM.

Moreover, from the simulation results of GISS-E2-H and GISS-E2-H-CC, it can also be found that

after considering the carbon cycle, GISS-E2-H-CC has no obvious improvement concerning the spatial
pattern of PIOAM, and the RMSEs in GISS-E2-H and GISS-E2-H-CC are basically the same. However,
the variance contribution of PIOAM in GISS-E2-H-CC is significantly improved compared to that in
GISS-E2-H, ranging from 36% to 44%, which is closer to the HadISST reanalysis. Unlike GISS-E2-H
and GISS-E2-H-CC, the variance contributions of PIOAM in GISS-E2-R and GISS-E2-R-CC are almost
the same, but with a smaller RMSE, the spatial pattern and standard deviation of PIOAM in GISS-E2-
R-CC are more consistent with the HadISST reanalysis than that in GISS-E2-R (see Fig. 2). Furthermore,
in the HadGEM2 model family, HadGEM2-CC considers the carbon cycle on the basis of HadGEM2-
AO. Compared to HadGEM2-AO, the RMSE of HadGEM2-CC is smaller, and the spatial type and
standard deviation of PIOAM are more consistent with the HadISST reanalsis. Again, the performance
of HadGEM2-CC effectively verifies the importance of the carbon cycle.

A comparison of the performance of MIROC-ESM and MIROC-ESM-CHEM revealed that the

chemical process also has an obvious influence on the simulation results of PIOAM. After the chemical
process is accounted for, the spatial standard deviation of PIOAM in MIROC-ESM-CHEM is closer to
the HadISST reanalysis than that in MIROC-ESM, but the RMSE in MIROC-ESM-CHEM is slightly
larger. The same is true for HadGEM2-E and HadGEM2-CC, indicating that the chemical process can
effectively improve the simulation effect of the standard deviation of PIOAM.

In addition, it can be found from the two models from the IPSL-CM5A family (IPSL-CM5A-LR

with low atmospheric resolution 95×96 and IPSL-CM5A-MR with medium atmospheric resolution
143×144) that atmospheric resolution also affects the PIOAM simulation results. The RMSE in IPSL-
CM5A-MR is slightly smaller than that in IPSL-CM5A-LR, and the spatial pattern of PIOAM is slightly
better. Moreover, the simulated standard deviation of PIOAM in IPSL-CM5A-MR is much closer to the
HadISST reanalysis than that in IPSL-CM5A-LR. This indicates that reasonably increasing the horizontal
resolution of atmospheric model may also improve the simulation effect on PIOAM.

**5. Conclusion and discussion**





Based on HadISST reanalysis from 1951 to 2005, the Pacific-Indian Ocean associated mode,
proposed by Yang and Li (2005) is evaluated for 21 CMIP5 models. This research provides a relatively
comprehensive evaluation of the spatial pattern, the interannual variation and the relationship with ENSO
and IOD of the PIOAM in the selected CMIP5 models. The main conclusions are as follows.
With a 47% contribution to total variance, the spatial pattern of PIOAM in the eastern equatorial
Pacific Ocean is a warm tongue, whereas there is negative SSTA in the western equatorial Pacific Ocean
that exhibits an obvious ENSO mode in the Pacific Ocean. In addition, the PIOAM presents an IOD
mode in the Indian Ocean. The variance contributions of PIOAM in almost all CMIP5 models are smaller
than that in the HadISST reanalysis. The simulation errors and differences among these models are
mainly concentrated in the Pacific Ocean, compared to the Indian Ocean, and a majority of models
overestimate the strength of PIOAM in the equatorial east Pacific and central Pacific. Although all these
models reproduce the spatial pattern of the positive SSTA in the eastern equatorial Pacific well, only one-
third of the models (CCSM4, CMCC-CM, CMCC-CMS, CNRM-CM5, FGOALS-s2, GFDL-ESM2M
and NorESM1-M) successfully simulate the ENSO mode with the east-west inverse phase in the Pacific
Ocean. In general, CCSM4, GFDL-ESM2M and CMCC-CMS have stronger capability to simulate the
PIOAM than the other models.
The PIOAM is very weak or not obvious in some years and has obvious seasonal and interannual
variations, as well as interdecadal variations. The simulated strengths of the PIOAM in the positive phase
are underestimated in most models; only less than one-fifth of the models (CCSM4, CMCC-CMS,
CNRM-CM5 and GFDL-ESM2M) are slightly stronger, and very close to the HadISST reanalysis,
especially CCSM4. The interannual variation of PIOAM in CCSM4, GISS-E2-R and FGOALS-s2 are
almost consistent with the HadISST reanalysis. Except that NorESM1-M overestimate the interannual
variation of PIOAM, the simulation results in most models are weak, especially MIROC-ESM. The
interannual variation of PIOAM in autumn can be measured by CCSM4, GISS-E2-R and FGOALS-s2.
The PIOAM can well reflect the activities of ENSO in the Pacific Ocean and IOD in the Indian Ocean
to a considerable extent with a close correlation to ENSO and IOD for the same period, as well as one
month in advance with ENSO.
Considering the carbon cycle or resolving the stratosphere can effectively improve the capability of
the model to simulate the PIOAM, a comparison of the performance of MIROC-ESM and MIROC-ESM-



CHEM revealed that the chemical process also has an obvious influence on the simulation results of
PIOAM. The chemical process can effectively improve the simulation effect of the standard deviation of
PIOAM. In addition, increasing the horizontal resolution of atmospheric model may improve the
simulation effect on PIOAM as well.

Yang et al. (2006) found that only considering the ENSO in the Pacific cannot entirely

explain the influence of SSTA on climate variation, and suggested that, to provide better
scientific explanation for short-term climate prediction, the PIOAM and its influence should be
considered and investigated. In addition, a review article by Cai et al. (2019) provides the first
comprehensive review and summary of the current research advances in the interaction between
the tropical Pacific-Indo-Atlantic climate systems, and they pointed out that an in-depth
understanding of the dynamic mechanisms of intertropical basin interactions is an important
way to improve the ability of seasonal to decadal climate prediction. Therefore, evaluating and
improving the capability of current climate models to simulate the PIOAM and even the tropical
Pacific-Indo-Atlantic climate systems are beneficial to obtain accurate climate change
predictions. In addition, improving the level of climate prediction is not only helpful to grasp
the changes in the ocean environment of the Pacific-Indian Ocean, but also propitious to
improve the ability of prediction and assessment of ocean waves and wind energy (Zheng and
Li, 2015; Zheng and Li, 2017).

**Data availability**. The CMIP5 data are available at https://esgf-node.llnl.gov/search/cmip5/. The sea
surface temperature are available at https://www.metoffice.gov.uk/hadobs/hadisst/data/download.html

**Author contributions**. Xin Li and Weilai Shi conceived the idea and designed the structure of this
paper; Minghao Yang performed the experiments; Minghao Yang, Chao Zhang and Jianqi Zhang
analyzed the data; Minghao Yang wrote the paper.

**Competing interests**. The author declares that they have no conflict of interest.

**Author Contributions**. This research was supported by National Natural Science Foundation of China

(4160501, 41490642, 41520104008).




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
