# Peer review of "The Pacific-Indian Ocean Associated Mode in CMIP5"

_Ocean Science, 2019_

## Referee Comment (RC1) · Anonymous Referee #1 · 29 Jul 2019

I have reviewed the manuscript 'The Pacifci-Indian Ocean Associated Mode in CMIP5 Models', by M. Yang et al. This manuscript discusses how the so-called Pacific-Indian Ocean Associated Mode is represented in CMIP5 models. But I do have a problem with this paper and even the topic. Based on the considerations below, I suggest to reject this manuscript.

Main points: 1. The only reference to the Pacifci-Indian Ocean Associated Mode I can find is related to a few publications by the authors themselfs. Essentially, what is meant by this Mode is the well known teleconnection between the Pacific (ENSO) and the Indian Ocean. Unfortunately, this study even fails to take the seasonality of this teleconnection into account. For example, in boreal winter the main mode of variability of the Indian ocean (the basin mode) is forced by ENSO, whereas in summer and

[Figure]

autumn the response of the Indian Ocean to ENSO projects onto the IOD (which is the focus of this study). However, this seasonality is important but not addressed at all. For example, this basin mode can be seen in Fig. 1, whereas the IOD response may be identified in Fig. 6. To not consider this seasonality makes the study essentially useless.

2. The ad-hoc definition in Eqs. 1,2,3 is not good enough. The common Indo-Pacific mode should be identified by an EOF analysis.

3. There is no in-depth analysis as to why the models do or do not represent the mode. Section 4 is pure speculation. The fact that some models including carbon cycle simulate the mode slightly better does not proof anything, if not supported by a large number of models, or by dedicated experiments.
* * *

---

## Referee Comment (RC2) · Ian Watterson (Referee) · 15 Aug 2019

General comments:

This is an interesting analysis of the tropical ocean surface temperatures from CMIP5 and HadISST. A 'mode' derived from the tropical Pacific-Indian domain has been denoted the PIOAM by some previous authors, mostly in Chinese journals. It seems worthwhile introducing the approach to this European one.

Here, it is shown that this Pacific-Indian mode (presumably obtained by principal component analysis) from 21 CMIP5 models has much in common with that from observations. There is a lengthy description of the differences among models. The mode is loosely linked to the IOD and ENSO, in section 3.1 of the study. The analysis in

section 3.2 focuses on alternative IOI and POI indices. A major problem is that a further index PIOAMI is then used as though it is the same as the first one (presumably PC1). Section 4 attempts to relate the differences between models to their differences in formulation. However, this is unconvincing, especially as there is no estimation of statistical uncertainty in results that have been obtained from a single 55-year period. Some conclusions are not well supported.

I initially thought the index might be linked to a 'Pacific-Indian Dipole' that I have used in analysing CMIP5 future climate simulations (see two recent references, below). However, the boxes used in that PID are a little shifted in longitude, so I expect there is only a weak relationship. Nevertheless, it might be worthwhile mentioning that alternative P-I index, and the shift.

The presentation in the paper is superficially quite good. However, there are many important details that are omitted, including in the captions. The 30 points listed below provide some guide to how the presentation needs to be improved. The major problem of having multiple indices, with no statistical uncertainty attached, will need to be overcome before final publication can be considered.

Significant points (at Line numbers):

1. L8-9 This needs to be a more helpful definition of how the PIOAM mode is defined, given that it is a rather new term. 2. L13 Why is HadISST referred to as a reanalysis? I don't think the Met Office does. 3. L46 Walker needs to have a capital W, as it is a person's name -in several places 4. L96 What CMIP5 simulations are used? Historical? 5. L101 Table 1 'oceanic resolution' might not be accurate given some have higher resolution in tropics. Is this the grid for the available ocean data? 6. L104 'Tropics' is normally considered bounded by 23 degrees latitude. Plot 1 shows 20S-20N. Which is used here? What are the longitudinal bounds? 7. L104 Is the analysis done on anomalies around a mean annual cycle? Is the data detrended? 8. L108 How is this 'mode' calculated? I presume it is a principal component / EOF analysis. The interval

of Fig 1, 0.003C makes this seem a very small amplitude. Could these EOF1 fields be scaled so they show the temperature anomaly for a 1 standard deviation of the index or PC1? Do the differences look the same? 9. L119 Error in longitudes for POI -should be 80W not 80E 10. L137 is rather late to state 'so-called'! 11. L138 What depth does heat refer to? 12. L138-140 This needs more discussion, perhaps earlier. What is the mathematical meaning? Should 'presents' be 'represents'? 13. L143 What is the statistical uncertainty of this analysis? There are only 55 years, or 20 ENSO cycles, perhaps. It would be good to obtain additional simulations from at least one model to give some indication. 14. L152 where is the ENSO mode shown? 15. L163-4 what does this mean? 16. L167 In the Fig 1 caption what is the % value? 17. L170 where is the IOD mode shown? 18. L192 It is confusing to have 'MME' of three models. What is MME at L193 and later in the paper? 19. L197 What is Fig. 2 actually showing? Comparisons of EOF1? What is REF? 20. L208 It would seem essential to compare the PC1 of PIOAM with this alternative PIOAMI index. If they are different, then the rest of the paper is misleading, whenever it compares 'PIOAM' with IOD, Nino34 etc. Is the PC1 more closely related to NINO34? 21. L225 What is 'autumn' for a tropical index? 'Boreal', and September-November perhaps? 22. L225-6 needs to be better written. Is this 1SD a criterion? 23. L239 The asymmetry in Fig 5 seems surprising. Does it indicate the index is not a 'normal' distribution? 24. L243 A composite, perhaps? 25. L255 Is this PIOAM or PIOAMI -here and later? 26. L256 Why is it interesting to show a standard deviation, when the IOI and POI are normalised (L120)? How does this impact the interannual autumn values? 27. L280 How is IOD defined here? Is it similar to IOI? 28. L316 This comparison needs to allow for statistical uncertainty, which should be considered early in the study. Would the results be the same if a different set of simulations is considered? 29. L325, L337 these statements are not convincingly proved. 30. L339. What is the chemical process?

Possible references: Watterson IG (2019) Influence of sea surface temperature on simulated future change in extreme rainfall in the Asia-Pacific. On-line, Asia-Pacific J. Atmos. Sci. doi 10.1007/s13143-019-00141-w Watterson IG (2019) Indices of climate

change based on patterns from CMIP5 models, and the range of projections. Climate Dynamics, 52, 2451-2466, doi 10.1007/s00382-018-4260-x

---

## Author Comment (AC1) · 18 Sep 2019

We thank the reviewer for the constructive comments that help improve the presentation of the original manuscript. Below are our point-to-point replies to the reviewer's comments (original comments are in italics):

Please see the PDF version in the Supplement.

Main points:

1. The only reference to the Pacific-Indian Ocean Associated Mode I can find is related to a few publications by the authors themselves. Essentially, what is meant by this Mode is the well-known teleconnection between the Pacific (ENSO) and the Indian Ocean. Unfortunately, this study even fails to take the seasonality of this teleconnection

into account. For example, in boreal winter the main mode of variability of the Indian ocean (the basin mode) is forced by ENSO, whereas in summer and autumn the response of the Indian Ocean to ENSO projects onto the IOD (which is the focus of this study). However, this seasonality is important but not addressed at all. For example, this basin mode can be seen in Fig. 1, whereas the IOD response may be identified in Fig. 6. To not consider this seasonality makes the study essentially useless.

Reply: Thank you for your comment very much. We followed the suggestion that seasonality is needed to be considered, and thank the specialist for precious advice. The Pacific-Indian Ocean associated mode (PIOAM), defined as the first dominant mode (empirical orthogonal function, EOF1) of SST anomalies in the Pacific-Indian Ocean between 20°S and 20°N. Figure below shows the pattern of SST anomalies over the Indo-Pacific Ocean in October 1982 and September 1997. It can be clearly seen that there are obvious warm tongues in the eastern equatorial Pacific Ocean, obvious positive SST anomalies in the northwest Indian Ocean, and obvious negative SST anomalies in the western equatorial Pacific Ocean and the eastern Indian Ocean. This is precisely the typical spatial pattern characteristics of the PIOAM. That is to say, the SST anomalies in the northwest Indian Ocean and the equatorial middle-east Pacific Ocean is opposite to the SST anomalies in the western equatorial Pacific Ocean and the east Indian Ocean. Compared with ENSO and IOD, the PIOAM has a broader spatial distribution.

Maps of SST anomalies for (a) October 1982 and (b) September 1997 from the HadISST dataset (unit: °C). The period from 1981 to 2005 is used to extract the monthly SST climatology.

However, is this spatial pattern of SST anomalies only a special case of a certain year, or is it stable? To answer this question, EOF analysis is performed on the SST anomalies of different seasons over Indo-Pacific Ocean (20°S-20°N, 40°E-80W°) from 1951 to 2005. All these first leading modes in Figure below are well separated from the remaining leading modes, based on the criteria of North et al. (1982), which means

less likely to be affected by statistical sampling errors. It can be found that the patterns of summer (June, July and August; Figure below b) and autumn (September, October and November; Figure below c) display the typical spatial distribution of the PIOAM, with the 46% and 61% contribution to total variance, respectively, while the spatial pattern of PIOAM is not so obvious in spring (March, April and May; Figure below a) and winter (December, January and February; Figure below d). In general, the PIOAM has stable structure and practical significance, especially in autumn.

Spatial patterns of the first leading mode of the (a) spring (March, April and May), (b) summer (June, July and August), (c) autumn (September, October and November) and (d) winter (December, January and February) averaged SST anomalies over Indo-Pacific Ocean (20°S-20°N, 40°E-80W°) calculated from HadISST dataset (unit: âĐČ). The numbers at the upper right corner of each panel indicate the percentage of variance explained by each season.

In addition, based on multi-variable empirical orthogonal functions, Chen and Cane (2008) and Chen (2011) also found this phenomenon and named it Indo-Pacific Tripole (IPT), which is considered to be an intrinsic mode in the tropical Indo-Pacific Ocean. In addition, Lian et al. (2014) used a conceptual model to discuss the development and physical mechanism of the IPT. Yang et al. (2006) found that the influences of the PIOAM and the ENSO mode on summer precipitation and climate in China were very different, and their numerical experiments also showed that the simulation results obtained by considering the PIOAM were more consistent with observation data. Therefore, evaluating and improving the capability of current climate models to simulate the PIOAM are beneficial to obtain accurate climate predictions.

Reference:

Chen, D., Cane, M. A.: El Niño prediction and predictability, J. Comput. Phys., 227, 3625-3640, doi: 10.1016/j.jcp.2007.05.014, 2008.

Chen, D.: Indo-Pacific Tripole: An intrinsic mode of tropical climate variability, Adv.

[Figure]

Geosci., 24, 1-18, doi: 10.1142/9789814355353_0001, 2011.

Lian, T., Chen, D. K., Tang, Y. M., Jin, B. G.: A theoretical investigation of the tropical Indo-Pacific tripole mode, Sci. China-Earth Sci., 57, 174-188, doi: 10.1007/s11430-013-4762-7, 2014.

North G. R., Bell, T. L., Cahalan, R. F., Moeng, F. J.: Sampling errors in the estimation of empirical orthogonal functions, Mon. Wea. Rev., 110, 699-706, doi: 10.1175/1520-0493(1982)110<0699:seiteo>2.0.co;2, 1982.

Yang, H., Jia, X. L. and Li, C. Y.: The tropical Pacific-Indian Ocean temperature anomaly mode and its effect, Chin. Sci. Bull., 51(23): 2878-2884, doi:10.1007/s11434-006-2199-5, 2006.

2. The ad-hoc definition in Eqs. 1,2,3 is not good enough. The common Indo-Pacific mode should be identified by an EOF analysis.

Reply: Thank you for your comment very much. It is customary to select the time coefficient (PC1) of the PIOAM as its index. It can be seen from the regression of the monthly SSTA onto the normalized PC1 (Figure below a) that the pattern in the Pacific Ocean is similar to ENSO, but positive SST anomalies occur throughout the Indian Ocean, which not matches the typical PIOAM spatial pattern. This is because the ENSO signals in the Pacific Ocean in PC1 are so strong that the signals of the IOD are not fully reflected. The correlation coefficient between PC1 and Niño3.4 index is as high as 0.95. However, obvious negative SST anomalies in the eastern Indian Ocean can be found in the regression map of the monthly SSTA based on the normalized PIOAMI (Figure below b) defined by Eqs. 1, 2, 3. The correlation coefficient between PIOAMI and Niño3.4 index is 0.68, indicating PIOAMI contains more Indian Ocean signals than PC1. In addition, the correlation coefficient between PC1 and PIOAMI is 0.70, which is far more than the confidence level of 99%. Therefore, PIOAMI can describe the mode well because of giving consideration to both the signals in the Pacific Ocean and the signals in the Indian Ocean.

Regressions of the monthly SSTA onto the normalized (a) PC1 and (b) PIOAMI for the period from 1951 to 2005 (unit: °C). The stippled areas for SSTA denote the 99% confidence levels.

3. There is no in-depth analysis as to why the models do or do not represent the mode. Section 4 is pure speculation. The fact that some models including carbon cycle simulate the mode slightly better does not proof anything, if not supported by a large number of models, or by dedicated experiments.

Reply: Thank you for your comment very much. We think your suggestion is correct and delete the Section 4 which is based on most of the speculation. The abstract and discussion are also be revised.

Please also note the supplement to this comment:
https://www.ocean-sci-discuss.net/os-2019-30/os-2019-30-AC1-supplement.zip

---

## Author Comment (AC2) · 18 Sep 2019

We thank the reviewer for the constructive comments that help improve the presentation of the original manuscript. Below are our point-to-point replies to the reviewer's comments (original comments are in italics):

***Please see the PDF version in the Supplement.

General comments: This is an interesting analysis of the tropical ocean surface temperatures from CMIP5 and HadISST. A 'mode' derived from the tropical Pacific-Indian domain has been denoted the PIOAM by some previous authors, mostly in Chinese journals. It seems worthwhile introducing the approach to this European one. Here, it is shown that this Pacific-Indian mode (presumably obtained by principal component

analysis) from 21 CMIP5 models has much in common with that from observations. There is a lengthy description of the differences among models. The mode is loosely linked to the IOD and ENSO, in section 3.1 of the study. The analysis in section 3.2 focuses on alternative IOI and POI indices. A major problem is that a further index PIOAMI is then used as though it is the same as the first one (presumably PC1). Section 4 attempts to relate the differences between models to their differences in formulation. However, this is unconvincing, especially as there is no estimation of statistical uncertainty in results that have been obtained from a single 55-year period. Some conclusions are not well supported. I initially thought the index might be linked to a 'Pacific-Indian Dipole' that I have used in analysing CMIP5 future climate simulations (see two recent references, below). However, the boxes used in that PID are a little shifted in longitude, so I expect there is only a weak relationship. Nevertheless, it might be worthwhile mentioning that alternative P-I index, and the shift.

The presentation in the paper is superficially quite good. However, there are many important details that are omitted, including in the captions. The 30 points listed below provide some guide to how the presentation needs to be improved. The major problem of having multiple indices, with no statistical uncertainty attached, will need to be overcome before final publication can be considered.

Significant points (at Line numbers):

1. L8-9 This needs to be a more helpful definition of how the PIOAM mode is defined, given that it is a rather new term.

Reply: Thank you for your suggestion. We have given the definition PIOAM in the abstract, which helps the reader to see our work more directly.

2. L13 Why is HadISST referred to as a reanalysis? I don't think the Met Office does.

Reply: Thank you for your comment. We have corrected it, replacing 'reanalysis' with 'dataset'.

3. L46 Walker needs to have a capital W, as it is a person's name -in several places

Reply: Thank you for your reminding. Done.

4. L96 What CMIP5 simulations are used? Historical?

Reply: Thank you for your reminding. Yes, it's historical, and we've added that.

5. L101 Table 1 'oceanic resolution' might not be accurate given some have higher resolution in tropics. Is this the grid for the available ocean data?

Reply: Thank you for your comment. We directly get the resolution of each model from the downloaded data, as showed below. In view of the resolution in tropical regions, we have supplemented it in the article.

6. L104 'Tropics' is normally considered bounded by 23 degrees latitude. Plot 1 shows 20S-20N. Which is used here? What are the longitudinal bounds?

Reply: Thank you for your comment very much. The 'Tropics' we are talking about here is to distinguish from the mid-latitudes. When it comes to computing, we use 20°S - 20°N. I'm sorry we didn't mention longitudinal bounds. It's 40°E -80°W. It has been added in the revised manuscript.

7. L104 Is the analysis done on anomalies around a mean annual cycle? Is the data detrended?

Reply: Thank you for your comment. We removed the annual cycle and the linear trend before performing the EOF on the tropical Pacific-Indian ocean SSTA. It has been supplemented in this article.

8. L108 How is this 'mode' calculated? I presume it is a principal component/EOF analysis. The interval of Fig 1, 0.003C makes this seem a very small amplitude. Could these EOF1 fields be scaled so they show the temperature anomaly for a 1 standard deviation of the index or PC1? Do the differences look the same?
Reply: Thank you for your comment very much. This mode is obtained by EOF analysis, and we have added additional explanation in the article. The Figure 1 with the interval 0.003°C is the direct results of EOF analysis. However, the amplitude of this pattern is relatively small. After scaling the patterns of this mode by normalizing the PC1 of each model and HadISST data set, the Figure 1 is redrawn with the interval 0.2°C. The distributions of differences are in accordance with previous results.

9. L119 Error in longitudes for POI -should be 80W not 80E

Reply: Thank you very much. It has been corrected.

10. L137 is rather late to state 'so-called'!

Reply: Thank you very much. It has been corrected.

11. L138 What depth does heat refer to?

Reply: Thank you for your comment. It refers to the surface, and it has been added.

12. L138-140 This needs more discussion, perhaps earlier. What is the mathematical meaning? Should 'presents' be 'represents'?

Reply: Thank you for your comment very much. This sentence does not correspond to what we are actually trying to convey. What we want to say is that the IOD is also like a meridional seesaw, and the text has been reworded.

13. L143 What is the statistical uncertainty of this analysis? There are only 55 years, or 20 ENSO cycles, perhaps. It would be good to obtain additional simulations from at least one model to give some indication.

Reply: Thank you for your comment very much. Because EOF analysis is used here to capture the pattern of PIOAM rather than composite analysis (Liu et al. 2013; Zhang and Sun 2014) or regression analysis (Jha et al. 2014; Chen et al. 2017; Chen et al. 2019), statistical significance test cannot be performed in Figure 1, which can also be seen in other studies (i.e. the North Pacific Oscillation in Wang et al. (2019), the Pacific

Decadal Oscillation in Lin et al. (2018), the interdecadal variability of SST in Lyu et al. (2016), the IOD and ENSO in Weller and Cai (2013)). Actually, the spatial correlation coefficients of the pattern of PIOAM between the HadISST and each CMIP5 model (in Figure 2) have significance at 99% confidence level. In addition, the first leading mode (PIOAM) is well separated from the remaining leading modes, based on the criteria of North et al. (1982), which means less likely to be affected by statistical sampling errors.

Reference:

Chen S, Wu R, Chen W, Song L. (2019) Performance of the CMIP5 models in simulating the Arctic Oscillation during boreal spring. Clim. Dyn. Doi: 10.1007/s00382-019-04792-3

Chen S, Chen W, Yu B. (2017) The influence of boreal spring Arctic Oscillation on the subsequent winter ENSO in CMIP5 models. Clim. Dyn. Doi: 10.1007/s00382-016-3243-z

Liu L, Xie SP, Zheng XT, Li T, Du Y, Huang G, Yu WD (2013) Indian Ocean variability in the CMIP5 multi-model ensemble: the zonal dipole mode. Clim. Dyn. Doi: 10.1007/s00382-013-2000-9

Lyu K, Zhang X, Church JA, Hu J. (2016) Evaluation of the interdecadal variability of sea surface temperature and sea level in the Pacific in CMIP3 and CMIP5 models. Int. J. Climatol. Doi: 10.1002/joc.4587

Lin R, Zheng F, Dong X. (2018) ENSO Frequency Asymmetry and the Pacific Decadal Oscillation in Observations and 19 CMIP5 Models. Adv. Atmos. Sci. Doi: 10.1007/s00376-017-7133-z

North G. R., Bell, T. L., Cahalan, R. F., Moeng, F. J. (1982) Sampling errors in the estimation of empirical orthogonal functions, Mon. Wea. Rev. Doi: 10.1175/1520-0493(1982)110<0699:seiteo>2.0.co;2

Weller E, Cai W. (2013) Asymmetry in the IOD and ENSO Teleconnection in a CMIP5

Model Ensemble and Its Relevance to Regional Rainfall. J. Clim. Doi: 10.1175/JCLI-D-12-00789.1

Wang X, Chen M, Wang C, Yeh SW, Tan W. (2019) Evaluation of performance of CMIP5 models in simulating the North Pacific Oscillation and El Niño Modoki. Clim. Dyn. Doi: 10.1007/s00382-018-4196-1

Zhang T, Sun DZ. (2014) ENSO Asymmetry in CMIP5 Models. J. Clim. Doi: 10.1175/JCLI-D-13-00454.1

14. L152 where is the ENSO mode shown?

Reply: Thank you for your comment. We mean that the spatial distribution of PIOAM in the Pacific Ocean is similar to ENSO model, which should be expressed as ENSO-like mode rather than ENSO mode. It has been revised.

15. L163-4 what does this mean?

Reply: Thank you for your comment. We want to briefly describe the PIOAM in MME. This sentence has been rewritten as 'MME better simulates the amplitude of PIOAM in the Indian Ocean than most these selected CMIP5 models with smaller simulation errors, but the amplitude in the equatorial Pacific are larger than that of the HadISST data set', which may not be that confusing caused by our inappropriate expressions.

16. L167 In the Fig 1 caption what is the % value?

Reply: Thank you for your comment. I'm sorry we overlooked it. The numbers at the upper right corner of each panel indicate the percentage of variance explained by each model. It has been added to the caption of Figure 1.

17. L170 where is the IOD mode shown?

Reply: Thank you for your comment. This is similar to the previous question which concerns ENSO mode. We want to express the pattern of PIOAM in the Indian Ocean is similar to IOD mode. We have replaced the IOD mode with IOD-like mode.

18. L192 It is confusing to have 'MME' of three models. What is MME at L193 and later in the paper?

Reply: Thank you for your comment. The MME in this article is multi-model ensemble and Its explanation is given in Section 3.1. The original L192 has been rephrased by deleting 'of these models' to avoid confusing expressions.

19. L197 What is Fig. 2 actually showing? Comparisons of EOF1? What is REF?

Reply: Thank you for your comment. The Taylor diagram presented in Figure 2 shows the ratio of the standard deviation calculated from simulation to that obtained in HadISST, spatial correlation coefficient and root mean square error (RMSE). The closer the point representing the model to REF, the better capability of the model. Actually, REF is a reference point. More detailed information of Taylor diagram can be found in Taylor (2001).

Reference:

Taylor, KE. (2001) Summarizing multiple aspects of model performance in a single diagram. J. Geophys. Res. Atmos. Doi:10.1029/2000jd900719.

20. L208 It would seem essential to compare the PC1 of PIOAM with this alternative PIOAMI index. If they are different, then the rest of the paper is misleading, whenever it compares 'PIOAM' with IOD, Nino34 etc. Is the PC1 more closely related to NINO34?

Reply: Thank you for your comment very much. In fact, since SST anomaly changes in the Indian Ocean are not as robust as those in the equatorial eastern Pacific Ocean, ENSO signals in the Pacific Ocean in PC1 are so strong that the signals of the IOD are not fully reflected, which can also be seen in the first picture (Figure a) below. However, PIOAMI can describe the mode well (Figure b), because this selected index takes into account both the signals in the Pacific Ocean and the signals in the Indian Ocean. In addition, the correlation coefficient between PC1 and PIOAMI is 0.70, which is far more than the confidence level of 99%. The correlation coefficients between Nino3.4

and PC1 and PIAOMI are 0.95 and 0.68, respectively, indicating that PC1 is indeed more closely related to Nino3.4.

Regressions of the monthly SSTA onto the normalized (a) PC1 and (b) PIOAMI for the period from 1951 to 2005. The stippled areas for SSTA denote the 99% confidence levels.

21. L225 What is 'autumn' for a tropical index? 'Boreal', and September-November perhaps?

Reply: Thank you for your comment. Autumn refers to September, October and November, which we have supplemented in the revised manuscript.

22. L225-6 needs to be better written. Is this 1SD a criterion?

Reply: Thank you for your suggestion. Yes, it is. we use one standard deviation as the criterion. This sentence has been rewritten.

23. L239 The asymmetry in Fig 5 seems surprising. Does it indicate the index is not a 'normal' distribution?

Reply: Thank you for your comment. We think the CMIP5 models have different performance to capture the PIOAM in different phase, which causes this obvious asymmetry.

24. L243 A composite, perhaps?

Reply: Thank you for your suggestion. Done.

25. L255 Is this PIOAM or PIOAMI -here and later?

Reply: Thank you for your comment. We apologize for the confusion. It should be PIOAMI when concerning other indices, like IOI, POI and Niño3.4 index. We have revised and unified it.

26. L256 Why is it interesting to show a standard deviation, when the IOI and POI are normalised (L120)? How does this impact the interannual autumn values?

Reply: Thank you for your comment very much. Here we calculated standard deviation to evaluate the ability of these CMIP5 models to simulate the interannual variation of the PIOAM. The closer the ratio is to 1, the better the ability to simulate interannual variation. Yes, it is. We normalized the IOI and POI in Section 2. Considering that PIOAMI is composed of IOI and POI, the interannual variations of IOI and POI may have an impact on PIOAMI, which may involve the internal process of PIOAM and needs further in-depth study.

27. L280 How is IOD defined here? Is it similar to IOI?

Reply: Thank you for your comment. We're sorry we didn't make it clear. Here the IOD is according to the definition of Saji et al. (1999), the difference in SSTA between the tropical western Indian Ocean (50°E-70°E, 10°S-10°N) and the tropical south-eastern Indian Ocean (90°E-110°E, 10°S-0), not it similar to IOI. It has been supplemented in the revised manuscript.

28. L316 This comparison needs to allow for statistical uncertainty, which should be considered early in the study. Would the results be the same if a different set of simulations is considered?

Reply: Thank you for your comment very much. We delete the Section 4 which is based on most of the speculation because the meaningful conclusions need to be supported by a large number of models, or by dedicated experiments. The abstract and discussion are also be revised.

29. L325, L337 these statements are not convincingly proved.

Reply: Thank you for your comment.

30. L339. What is the chemical process?

Reply: Thank you for your comment.

Possible references:
Watterson IG (2019) Influence of sea surface temperature on simulated future change in extreme rainfall in the Asia-Pacific. On-line, Asia-Pacific J. Atmos. Sci. doi 10.1007/s13143-019-00141-w

Watterson IG (2019) Indices of climate change based on patterns from CMIP5 models, and the range of projections. Climate Dynamics, 52, 2451-2466, doi 10.1007/s00382-018-4260-x

Please also note the supplement to this comment:
https://www.ocean-sci-discuss.net/os-2019-30/os-2019-30-AC2-supplement.zip
* * *

---

## Author Response (AR3)

We thank the reviewer again for the constructive comments that help improve the presentation of the original manuscript. Below are our point-to-point replies to the reviewer's comments (original comments are in italics):

*I thank the authors for their major revision of the paper and responses to the points made. It is much more suitable for final publication. The first reviewer's advice regarding seasonality has been very helpful, and the revision documents how the pattern of the EOF1 differs with season. From (new line) L137 and 3.2.1, it seems that the authors prefer a pattern that has both signs in the Indian Ocean, as in JJA and SON. (Regarding my point 21, this is winter and spring, on my side of the Equator, but the Journal's practice on this should be followed.) This prompts a second index, after PC1, which was found to better relate to Asia (L119). Unfortunately, the revision has largely disregarded my concerns (general comment, points 20, 25) about how these two indices for 'PIOAM' are presented. There is no mention of two indices in the abstract. The conclusions describe results from each form, but without acknowledging this.*

*Some further revision is needed, I think. If both indices are used, then this might be guided by previous papers. It seems the all-month PC1 will need to be retained, given its use in Fig. 3. The second index definition might be further justified as a practical one defined with boxes, following the approach of IOD, NINO34, etc. Are the two indices better correlated in JJA and SON? Would it be interesting to extend Fig 6 to four seasons?*

**Reply: Thank you very much for your comments and suggestions. We attach great importance to and seriously consider your comments, but we are sorry we didn't make the point-by-point reply clear. For points 20, we replied that the PC1 of PIOAM is compared with the alternative PIOAMI (the second index) by the Figure 6 in the revised manuscript, and the correlation coefficients between Nino3.4 and PC1 and PIAOMI are 0.95 and 0.68, respectively, indicating that PC1 is indeed more closely related to Nino3.4. For points 25, we apologize for misunderstanding the reviewer's concerns.**

**For Fig. 3, EOF analysis is used to investigate the spatial pattern in HadISST dataset and CMIP5 models, without using PC1. In fact, we don't use PC1 anywhere except in 3.2.1 when comparing PIOAMI (the second index) with PC1. We have explained it in the article, please see lines 274-277. The correlation coefficients between the two indices in MAM, JJA, SON and DJF are 0.45, 0.76, 0.87 and 0.71, respectively, which indicates that the two indices are better correlated in JJA and SON. In addition, the figures below show the regressions of the SSTA onto the normalized PC1 and PIOAMI in four seasons. It can also be found that the spatial patterns associated with PIOAMI (e-h) are closer to the typical spatial pattern of the PIOAM than that associated with PC1 (a-d). This part has been added to the revised manuscript, please see lines 272-281. Just as several indices can describe ENSO, the PC1 and the so-called PIOAMI can also describe PIOAM. However, in the present study, we believe that the so-called PIOAMI can better represent the PIOAM than the PC1. Therefore, we chose to use the so-called PIOAMI to investigate PIOAM in the following studies, instead of using the both indices. Furthermore, we emphasize the two indices in the abstract, please see lines 16-18.**

[Figure]

Regressions of the (a, e) MAM, (b, f) JJA, (c, g) SON and (d, h) DJF SSTA onto the normalized (a-d) PC1 and (e-h) PIOAMI (unit: °C). The stippled areas for SSTA denote the 99% confidence levels.